# Anthocyanins Reduce Cell Invasion and Migration through Akt/mTOR Downregulation and Apoptosis Activation in Triple-Negative Breast Cancer Cells: A Systematic Review and Meta-Analysis

**DOI:** 10.3390/cancers15082300

**Published:** 2023-04-14

**Authors:** Ana Carolina Silveira Rabelo, Caroline de Aquino Guerreiro, Vivian Izumi Shinzato, Thomas Prates Ong, Giuliana Noratto

**Affiliations:** 1Department of Food and Experimental Nutrition, Faculty of Pharmaceutical Sciences, University of São Paulo, São Paulo 05508-270, Brazil; 2Food Research Center (FoRC)—Food Research Center, University of São Paulo, São Paulo 05508-270, Brazil; 3Department of Nutrition and Food Science, Texas A&M University, College Station, TX 77843, USA

**Keywords:** chemoprevention, black rice, dark sweet cherry, cyanidin-3-glucoside, bioactive compounds, natural products

## Abstract

**Simple Summary:**

Breast cancer is the second most common cancer and the most commonly occurring cancer among women in the world. It is estimated that more than 2 million cases emerged in 2020 and that this number will increase even more in the coming years. More specifically, triple-negative breast cancer (TNBC) accounts for 10–20% of invasive breast cancers and represents a consistent subgroup of breast cancers with heterogeneous clinical presentation, behavior, pathology, and response to treatment. The standard therapy for TNBC patients consists of chemotherapy; however, about 50% of patients develop drug resistance, promoting poor overall survival. Thus, studies have focused on discovering complementary therapeutic options, including the use of anthocyanins. In this sense, the systematic review and meta-analysis can be seen as a powerful tool in the compilation and statistical analysis of data from primary articles to prove the effects promoted by anthocyanins in TNBC cells suggested in the literature.

**Abstract:**

Background: Studies have suggested the chemopreventive effects of anthocyanins on breast cancer carcinogenesis. This systematic review and meta-analysis aimed to evaluate the effect of anthocyanins on triple-negative breast cancer cells (TNBC) cultured in vitro. Methods: We searched for all relevant studies that evaluated the mechanisms of migration, invasion, Akt/mTOR and MAPK pathways, and apoptosis, using PubMed and Scopus. Means and standard deviation were used, and a randomized effects model was applied, with a confidence interval of 95%. Statistical heterogeneity between studies was assessed using the Chi2 test and I2 statistics. All analyses were performed using RevMan software (version 5.4). Results: Eleven studies were included in the systematic review and ten in the meta-analysis, where the roles of anthocyanin-enriched extract or cyanidin-3-O-glucoside (C-3-O-G) on MDA-MB-231 and MDA-MB-453 cells were investigated. Discussion: There was a significant reduction in invasion (mean difference: −98.64; 95% CI: −153.98, −43.3; *p* ˂ 0.00001) and migration (mean difference: −90.13; 95% CI: −130.57, −49.68; *p* ˂ 0.00001) in TNBC cells after anthocyanins treatment. Anthocyanins also downregulated Akt (mean difference: −0.63; 95% CI: −0.70, −0.57; *p* ˂ 0.00001) and mTOR (mean difference: −0.93; 95% CI: −1.58, −0.29; *p* = 0.005), while JNK (mean difference: −0.06; 95% CI: −1.21, 1.09; *p* = 0.92) and p38 (mean difference: 0.05; 95% CI: −1.32, 1.41; *p* = 0.95) were not modulated. There was also an increase in cleaved caspase-3 (mean difference: 1.13; 95% CI: 0.11, 2.16; *p* = 0.03), cleaved caspase-8 (mean difference: 1.64; 95% CI: 0.05, 3.22; *p* = 0.04), and cleaved PARP (mean difference: 0.93; 95% CI: 0.54, 1.32). Although the difference between control and anthocyanin groups was not significant regarding apoptosis rate (mean difference: 3.63; 95% CI: −2.88, 10.14; *p* = 0.27), the analysis between subgroups showed that anthocyanins are more favorable in inducing overall apoptosis (*p* ˂ 0.00001). Conclusion: The results show that anthocyanins hold promise in fighting against TNBC, but their effects should not be generalized. In addition, further primary studies should be conducted so that more accurate conclusions can be drawn.

## 1. Introduction

Breast cancer is a disease that compromises women’s health worldwide. According to the World Health Organization, 2.3 million women were diagnosed with breast cancer in 2020, making it the most prevalent cancer in the world [1]. Breast cancer is a highly heterogeneous disease, characterized by distinct genetic, molecular, and histological characteristics. As a result, each case of breast cancer is unique and differs from one another, leading to varying clinical outcomes and responses to treatment [2]. One of the most discussed is triple-negative breast cancer (TNBC), characterized by the absence of the expression of hormone receptors for estrogen, progesterone, and the human epidermal growth factor protein [2]. According to the American Cancer Society [3], TNBC accounts for 10–15% of all diagnosed breast cancers and remains a challenge for physicians and researchers given its relevance, high incidence, aggressiveness, limited treatment options, and worse prognosis.

Therefore, the search for natural compounds present in foods that can help in the prevention and treatment of breast cancer is proposed. In this context, food-derived anthocyanins have shown the potential to inhibit TNBC in different research studies, including in pre-clinical research [4,5]. Anthocyanins are water-soluble substances that belong to the class of polyphenolic compounds [6] responsible for the blue, red, and purple colors of fruits, vegetables, and plants [7]. Alongside other phenolic compounds, anthocyanins have anticancer chemopreventive effects associated with an increase in apoptosis, a decrease in cell proliferation, as well as cell cycle arrest [8,9]. Due to its wider distribution in food, anthocyanins such as the cyanidin-3-glucoside (C-3-O-G) (see structure in Figure 1) are the most studied among other flavonoid compounds [10].

Anthocyanins seem to target pathways involved in drug resistance [11], proliferation, angiogenesis, and metastasis, specifically in breast cancer [12,13]. Anthocyanin’s antioxidant properties are frequently involved in reducing the cell viability of breast cancer cell lines, including the TNBC cells [14]. This may imply a possible subtype-specific mechanism [5].

Considering the relevance of TNBC and studies reporting treatment with anthocyanins at different doses and administration modes, this study aimed to evaluate their role when used as extracts enriched in anthocyanins or isolated anthocyanin (C-3-O-G) in TNBC through a systematic review and meta-analysis.

## 2. Materials and Methods

### 2.1. Search Strategy

The systematic review and meta-analysis were conducted under the Preferred Reporting Items for Systematic Reviews and Meta-Analyses (PRISMA) guidelines [15]. The literature search was conducted by three researchers (A.C.S.R., C.A.G. and V.I.S.) independently using the two major databases, PubMed and Scopus, before June 2022. The search used the terms breast cancer AND (anthocyanins OR cyanidin-3-glucoside).

### 2.2. Study Selection

Initially, the researchers screened the titles and abstracts of all papers to exclude studies that did not meet the inclusion criteria. The full-text versions of all potentially relevant studies were screened, and disagreements were resolved through discussions until consensus was achieved.

Articles were considered eligible for the analysis if they met the following inclusion criteria: (1) evaluation of the biological effects of anthocyanins on TNBC cultured in vitro; (2) data related to wound healing, cell invasion, cellular pathways related to migration and invasion processes (such as Akt/mTOR, MAPK, epithelial–mesenchymal transition markers), and apoptosis induced by anthocyanins; (3) extracts enriched with anthocyanin or isolated anthocyanin (cyanidin-3-glucoside).

Exclusion criteria were (1) review, book, book chapter, letters, or commentaries; (2) tests performed in cell lines other than TNBC; (3) in silico analysis only; (4) in vivo assays only; (5) isolated anthocyanin other than cyanidin-3-O-glucoside; (6) full-length articles in languages other than English; (7) none of the above-mentioned assays; (8) assays that combined anthocyanins with chemotherapies or other drugs; (9) tests that evaluated breast cancer associated with other conditions (such as menopause, alcohol abuse, etc.).

### 2.3. Data Extraction

Data from articles that met the inclusion criteria were extracted from a standard spreadsheet. Extracted data included the study description (authors, year, and analyzed parameters), characteristics of treatment (source of anthocyanin-enriched extract or isolation, dosage, duration), characteristics of the cells (cell line), and the reported results (mean and standard deviation; when numerical values of results were not reported in tables, graphs were used, and data were extracted using the WebPlotDigitizer tool [16]). Authors were contacted to share their data for inclusion in the meta-analysis when relevant data were not shown in the manuscript. Studies were excluded if authors did not respond.

When studies used more than a single anthocyanin dose, the Review Manager software calculator tool (The Cochrane Collaboration, The Nordic Cochrane Centre, Copenhagen, Denmark) was used to combine the means and standard deviations of each concentration.

### 2.4. Statistical Analysis

The statistical heterogeneity among studies was assessed by the Chi2 test and I2 statistics. The meta-analysis was carried out using the Review Manager (RevMan 5.3) software. The anthocyanin-treated (enriched extract or isolated) TNBC cells were compared to untreated controls in regard to migration, invasion, and apoptosis. A random statistical model was used for the meta-analysis of all selected parameters, with its 95% confidence intervals (CI), where a *p*-value <0.05 was considered statistically significant.

## 3. Results

### 3.1. Characteristics of the Included Studies

The initial research strategy yielded 354 records (94 from PubMed and 260 from Scopus). Of the 354 items, 87 were duplicates and were excluded. After an initial screening of titles and abstracts, a further 252 elements [Review/book/letter (n = 122); analysis of other cancer types (n = 48); in silico analysis only (n = 5); in vivo assays only (n = 2); did not involve anthocyanin-enriched extract (n = 39); others (n = 36)] were excluded. Sixteen full-text articles were retrieved and assessed, but only eleven studies met the inclusion criteria after a full review [14,17,18,19,20,21,22,23,24,25,26]. One of the studies was excluded because it did not contain the number of replicates evaluated per experiment [21]. Thus, ten studies were included in the meta-analysis. The article selection process is summarized in Figure 2.

The characteristics of all included studies are presented in Table 1. The studies included in the final analysis investigated anthocyanins from Black Rice (BRACs) (*Oryza sativa* L.) (n = 4), Dark Sweet Cherry (*Prunus avium)* (ACN) (n = 2), *Eugenia jambolana* (n = 1), and isolated Cyanidin-3-O-glucoside (C-3-O-G) (n = 4). Of these studies, six evaluated the processes of invasion and migration, four papers analyzed cell growth signaling pathways (Akt, mTOR, JNK, and p38), and five studies evaluated apoptosis (caspases, PARP, and apoptotic rate) induced by anthocyanins in MDA-MB-231 and MDA-MB-453 cells.

### 3.2. Anthocyanins Reduced Invasion and Migration of MDA-MB-231 and MDA-MB-453 Cells

To investigate the role of anthocyanins in the processes of invasion and migration, a meta-analysis was performed with six studies that met the inclusion criteria and performed the wounding healing and cell motility assays [18,19,22,23,24,26]. The meta-analysis was carried out in subgroups, with four studies conducted using the anthocyanin-enriched extract [18,19,22,26], and two studies that evaluated the isolated anthocyanin C-3-O-G [23,24]. There was high statistical heterogeneity between the studies of both invasion (*p* ˂ 0.00001; I2 = 99%) and migration processes (*p* ˂ 0.00001; I2 = 96%) (Figure 2 and Figure 3). Overall, both the anthocyanins-enriched extract (mean difference: −34.73; 95% CI: −63.91, −5.55; *p* = 0.02) and the isolated C-3-O-G (mean difference: −214.97; 95% CI: −229.88, −200.05; *p* ˂ 0.00001) inhibited the TNBC cell invasion process when analyzed alone or in subgroups (mean difference: −98.64; 95% CI: −153.98, −43.3; *p* ˂ 0.00001) (Figure 3). Regarding migration, the anthocyanin-enriched extract was shown to reduce this process (mean difference: −60.45; 95% CI: −68.71, −52.19; *p* ˂ 0.00001), but isolated C-3-O-G did not have the same effect (mean difference: −170.30; 95% CI: −417.59, 76.98; *p* = 0.18). However, when subgroup analysis was considered, both anthocyanin-enriched extract and isolated C-3-O-G reduced migration (mean difference: −90.13; 95% CI: −130.57, −49.68; *p* ˂ 0.00001) (Figure 4).

### 3.3. Anthocyanins Target the Akt/mTOR Pathway in MDA-MB-231 and MDA-MB-453 Cells

Several cell signaling pathways, such as Akt/mTOR and MAPK, are involved in the processes of growth, survival, migration, and invasion of tumor cells. Therefore, research has focused on these pathways as a possible therapeutic target for TNBC. As shown in Figure 4, a meta-analysis was conducted to evaluate the modulatory potential of ACN on Akt/mTOR and MAPK (JNK and p38) pathways. Four studies matched the inclusion criteria, two evaluating Akt phosphorylation [20,26], one analyzing mRNA levels and mTOR expression [25], two evaluating JNK [19,26], and two analyzing p38 [25,26].

The studies showed no statistical heterogeneity for the Akt analysis (*p* = 0.69; I2 = 0%) but showed high heterogeneity among studies that evaluated mTOR (*p* = 0.0003; I2 = 92%), JNK (*p* ˂ 0.00001; I2 = 98%), and p38 (*p* ˂ 0.00001; I2 = 99%). Overall, Akt (mean difference: −0.63; 95% CI: −0.70, −0.57; *p* ˂ 0.00001) and mTOR (mean difference: −0.93; 95% CI: −1.58, −0.29; *p*= 0.005) were downregulated by anthocyanins, while JNK (mean difference: −0.06; 95% CI: −1.21, 1.09; *p* = 0.92) and p38 (mean difference: 0.05; 95% CI: −1.32, 1.41; *p* = 0.95) were not modulated by them (Figure 5).

### 3.4. Anthocyanins Induce Apoptosis in MDA-MB-231 and MDA-MB-453 Cells

Apoptosis induction by natural compounds in cancer cells is a well-known mechanism. In this context, a meta-analysis was performed to evaluate parameters related to programmed cell death, i.e., cleaved caspases 3 and 8, cleaved PARP, and apoptosis rate. Five studies matched the inclusion criteria [14,17,20,25,26].

The meta-analysis showed a high heterogeneity between studies related to cleaved caspase-3 (*p* ˂ 0.00001; I^2^ = 94%), cleaved caspase-8 (*p* ˂ 0.00001; I^2^ = 98%), cleaved PARP (*p* ˂ 0.00001; I^2^ = 93%), and apoptosis rate (*p* = 0.0006; I^2^ = 92%). The mean differences and CI in relation to cleaved caspase-3 (mean difference: 1.13; 95% CI: 0.11, 2.16; *p* = 0.03), cleaved caspase-8 (mean difference: 1.64; 95% CI: 0.05, 3.22; *p* = 0.04), and cleaved PARP (mean difference: 0.93; 95% CI: 0.54, 1.32; *p* ˂ 0.00001) indicated the anthocyanins group as being more favorable in inducing apoptosis compared to the control. Although the difference between control and anthocyanin groups was not significant regarding apoptosis rate (mean difference: 3.63; 95% CI: −2.88, 10.14; *p* = 0.27), the analysis between subgroups showed that anthocyanins are more favorable in inducing apoptosis than the control group (*p* ˂ 0.00001) (Figure 6).

## 4. Discussion

The collection of empirical evidence according to pre-defined eligibility criteria to answer a defined research problem is called a systematic review, while the application of statistical methods in the synthesis of such results is a meta-analysis [27]. This systematic review and meta-analysis are, to the best of our knowledge, the first to investigate the effect of anthocyanin treatment on TNBC, which is associated with poor prognosis among breast cancer patients. Findings showed that anthocyanins, both in enriched extracts or isolated, significantly decreased the invasion and migration of MDA-MB-231 and MDA-MB-453 cells. It has also been shown that anthocyanins primarily target the Akt/mTOR pathway but not the MAPK stress-activated and pro-apoptotic proteins JNK and p38. In addition, the apoptotic process was more favorable to the anthocyanins group than the control group cells.

The HER2+ as well as the TNBC subtypes are aggressive cancers that grow rapidly, with a high probability of spreading at the time of diagnosis. Moreover, these breast cancer subtypes have a high probability of post-treatment recurrence [3]. Due to the poor prognosis of metastatic breast cancer, understanding the underlying mechanisms to help in preventing its spread is a priority [28]. In this context, anthocyanins have been extensively studied.

In this review, according to the predefined inclusion criteria, 10 articles were selected from a wide range of in vitro studies regarding the action of anthocyanins on the TNBC MDA-MB-231 cells and the HER2+ MDA-MB-453 cells. Regarding the invasion and migration mechanisms, the data meta-analysis showed that anthocyanin-enriched extract and isolated C-3-O-G reduced both cell invasion and migration. A similar result was obtained by Chen et al. [19] who investigated the effect of black rice anthocyanins (BRACs) on cell migration and invasion in MDA-MB-453 cells. The 200 µg/mL BRACs treatment for 24 h reduced invasion and migration by approximately 41% and 45%, respectively. In addition, Luo et al. [18] and Zhou et al. [22] reported that 200 μg/mL BRACs treatment for 24 h was also effective in reducing cell invasion and migration of MDA-MB-453 cells. Although treatment with ACN (70 µg C3G equivalent/mL) reduced the invasion process by 57.8% in MDA-MB-453 cells, there was no statistical difference. However, the same extract was statistically effective in reducing the percentage of cells that invaded the wounded area down to 44% [26]. In addition to the potential of anthocyanin-enriched extracts, isolated C-3-O-G was also effective in reducing the processes of invasion and migration in MDA-MB-231 and MDA-MB-453 cells [23,24].

The PI3K/Akt/mTOR pathway is frequently involved in physiological and abnormal cell growth and proliferation [29], in addition to cell survival, motility, and immune response [30]. Briefly, the PI3K protein is phosphorylated and activated by the tyrosine kinase receptor (RTK). Then, PI3K phosphorylates PIP2, resulting in PIP3 production. Activated PIP3 mediates Akt activation, which, in turn, is responsible for the inhibition of the Ras homolog enriched in brain (RHEB) GTPase activity of the TSC1/2 complex. Once activated, Rheb stimulates the mTOR complex 1 to phosphorylate p70S6 and 4EBP1 proteins, promoting the dysregulation of protein synthesis and cell survival [31]. About 60% of all breast cancer tumors have the Akt/mTOR pathway activated due to genetic alterations, which contributes to therapy resistance [32,33]. Therefore, blocking the Akt/mTOR pathway has been linked to reduced tumor growth and increased patient survival [34]. Furthermore, additional anthocyanins present in fruits and vegetables have been found to contribute to the downregulation of Akt expression in breast cancer cells. For instance, a study revealed that anthocyanins extracted from *Vitis coignetiae* Pulliat (also known as Meoru in Korea) promoted the downregulation of Akt expression in MCF-7 breast cancer cells. In addition, the study also demonstrated that anthocyanins enhanced cisplatin sensitivity by inhibiting the Akt and NF-κB activity of breast cancer cells that show intrinsic resistance to chemotherapy [11]. Another study demonstrated that treatment of breast cancer cells with delphinidin-3-glucoside, an anthocyanin commonly found in pigmented fruits and vegetables, resulted in reduced Akt expression, leading to a subsequent decrease in the expression of HOTAIR, an oncogene linked to the development and metastasis of various cancers, including breast cancer [35].

When the meta-analysis was carried out, it became evident that anthocyanins target the Akt/mTOR pathway to reduce tumor cell growth, migration, and invasion processes. As demonstrated, both C-3-O-G and ACN have been shown to decrease the expression of p-AKT in MDA-MB-231 and MDA-MB-453 cells, respectively [20,25]. Akt mRNA levels were also downregulated by ACN (0.4-fold of untreated control) in MDA-MB-453 cells [25]. Consistently, ACN also reduced the expression and mRNA levels of Akt-downstream mTOR, contributing to the reduction in cell invasion and migration [25].

The crosstalk between the Akt/mTOR and MAPK/MEK/ERK pathways that drive tumor growth and cell spread is well known. Therefore, the modulation of these pathways plays an important role as a therapeutic approach to treating TNBC [36]. Studies have shown that the anthocyanin treatments modulated MAPK family members, such as JNK, p38, and ERK1/2 [19,26]. Chen et al. [19] demonstrated that BRACs treatment promoted a reduction in JNK mRNA levels; however, in the study conducted by Layosa et al. (2021) [26], ACN promoted the upregulation of JNK and p38, which culminated in the activation of apoptosis and reduced migration of tumor cells. These data suggest that anthocyanins may act at the transcription and post-transcription levels, depending on their source; however, the meta-analysis of these results showed that anthocyanins do not promote JNK or p38 modulation.

The epithelial–mesenchymal transition (EMT) process has also been described as an important factor in the malignancy and spread of tumor cells. Treatments with anthocyanin-enriched extracts and isolated C-3-O-G were effective in reducing EMT markers. Liang et al. [23] demonstrated that non-treated MDA-MB-231 cells displayed fibroblastoid and mesenchymal phenotypes, but after C-3-O-G treatment, cells showed epithelial-like morphology. Furthermore, cells showed an increase in E-cadherin and Zonula occludens-1 (ZO-1) and a reduction in Vimentin and EMT-associated transcription factors Snail1, and Snail2 after C-3-O-G, indicating its activity in reversing EMT in breast cancer cells. Hui et al. [14] also demonstrated that BRACs were able to reduce metalloproteinases 2 and 9, which are critical proteins in extracellular matrix degradation, allowing cells to migrate from the initial site to distant organs. It was also shown that BRACs reduced the activation of Src and FAK, two transcription factors related to EMT activation [22]. Other studies have also shown that anthocyanins target EMT to reduce TNBC cell migration and invasion [19,24]. Even though EMT plays a key role in TNBC cells migration and invasion processes, there were not enough primary studies to perform the meta-analysis and provide conclusive evidence regarding the role of anthocyanins in EMT.

The induction of apoptosis or programmed cell death is considered an effective mechanism against cancer cells. During the activation of this process, cell shrinkage, cytoplasm density, and more tightly-packed organelles are visualized, in addition to pyknosis. Molecularly, the cleavage of caspase-3 appears to be essential for inducing the apoptotic process that results in DNA fragmentation, protein degradation, and formation of apoptotic bodies [37]. Apoptosis appears to be induced by various natural products that seem to be effective in the fight against breast cancer [38]. Treatment with 200 µg/mL of *Eugenia jambolana* anthocyanins increased the apoptotic rate in MDA-MB-231 cells [17]. Cho et al. [21] also demonstrated that C-3-O-G isolated from mulberry (*Morus alba* L.) promoted an increase in the small size of oligonucleosomal fragments and the levels of cleaved caspase-3 per caspase-3, in a dose-dependent manner. In addition, C-3-O-G downregulated levels of Bcl-2, a regulator of mitochondrial outer membrane permeability that prevents apoptosis. Apoptosis also appears to be an important mechanism for the antitumor effect of BRACs, as 150 and 200 µg/mL of this extract led to a loss of mitochondrial membrane potential, with consequent release of cytochrome C into the cytosol. These events led to activated caspase-3 and -9, and induced remarkable cleavage of PARP in MDA-MB-453 cells. However, Hui et al. (2010) did not observe significant cleavage of caspase-8 in MDA-MB-231 cells [14].

ACN did not modulate mRNA levels of caspase-3, -8, -9, and Bax/Bcl-2 [25], but it upregulated cleaved caspase-8 and Bax/Bcl-2 rate, alongside promoting increased PARP cleavage, a hallmark of apoptosis [26]. These data strongly suggest that ACN modulates the expression of apoptotic molecules at the post-transcriptional levels. The treatment of MDA-MB-231 cells with C-3-O-G did not promote changes in mitochondrial function, but it increased cleaved-caspase-3 and 8, suggesting that C-3-O-G activates apoptosis independently of the intrinsic pathway [20]. Interestingly, this meta-analysis supported the data available in the literature and showed that anthocyanins induced apoptosis.

## 5. Conclusions and Future Directions

This data meta-analysis indicated that anthocyanin-enriched extracts and isolated C-3-O-G were able to reduce both cell migration and invasion by mechanisms likely involving the inhibition of the Akt/mTOR pathway and induction of apoptosis. These findings show that anthocyanins hold promise in fighting against TNBC, but their effects should not be generalized. This work has some limitations, including the small number of selected studies that fit the pre-established criteria, the high heterogeneity present in the data analyzed, and the limited availability of studies with anthocyanin-enriched extracts or isolated anthocyanins in TNBC. Thus, caution should be practiced when interpreting the overall results. The need for more primary studies investigating the role of anthocyanins in TNBC is imperative to support future breast cancer clinical studies in humans.

## Figures and Tables

**Figure 1 cancers-15-02300-f001:**
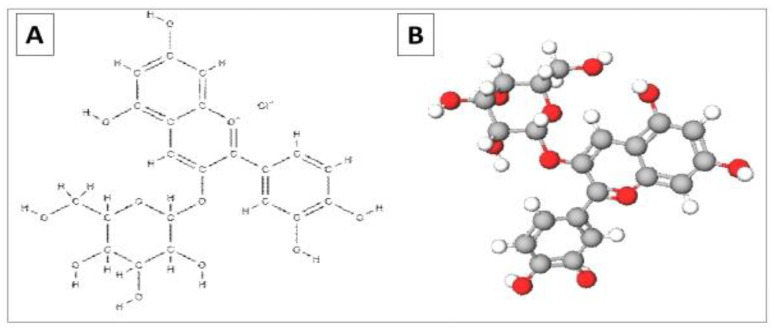
Structure of anthocyanin (cyanidin-3-glucoside). (**A**): 2D structure; (**B**): 3D structure. Created with https://molview.org/ (Acessed on 12 March 2023).

**Figure 2 cancers-15-02300-f002:**
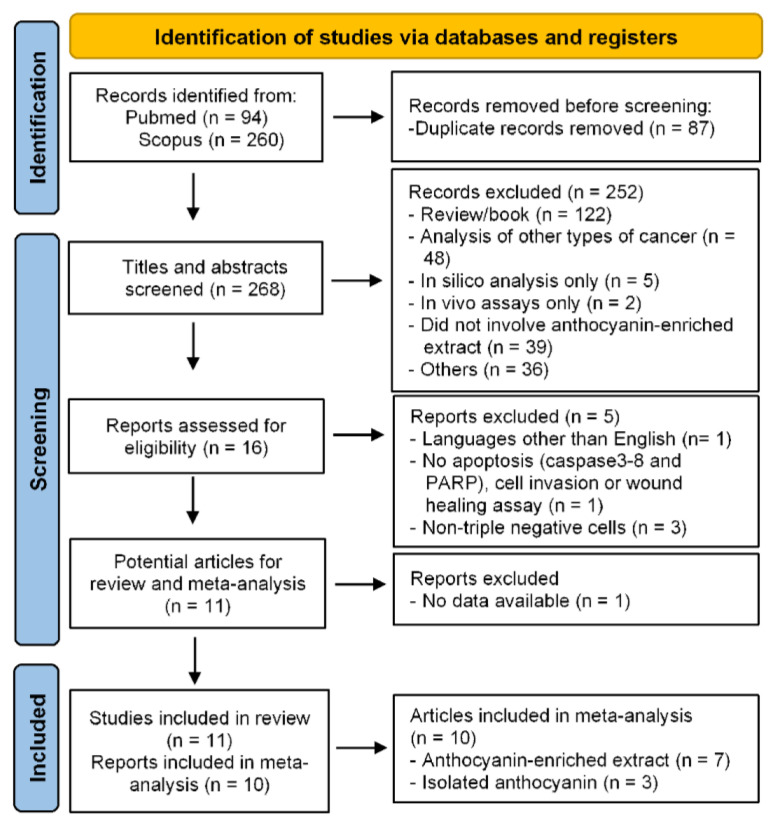
Flowchart showing the study selection process following the Preferred Reporting Items for Systematic Reviews and Meta-Analyses.

**Figure 3 cancers-15-02300-f003:**
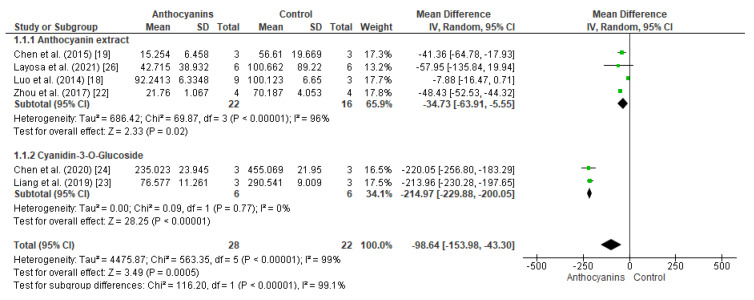
Forest plot of the meta-analysis comparing the anthocyanin-treated TNBC cell line versus the untreated controls in invasion. The analysis was performed in subgroups, with four studies conducted with the anthocyanin-enriched extract and two studies evaluating isolated anthocyanin (cyanidin-3-O-glucoside). IV: inverse variance based on the mean difference; CI: confidence interval. The green box in the middle of each horizontal line represents the point estimate of the effect for each study (the larger the size of the green box, the greater the weight of the study). The black diamond represents the effect estimate of all meta-analysis studies. The position of the diamond indicates which group was significant in modulating the parameter (e.g., if the diamond is on the left side, it means that the anthocyanin group modulated the parameter. However, if the diamond touches the central vertical line, it means that there is no difference between control and anthocyanin groups) [18,19,22,23,24,26].

**Figure 4 cancers-15-02300-f004:**
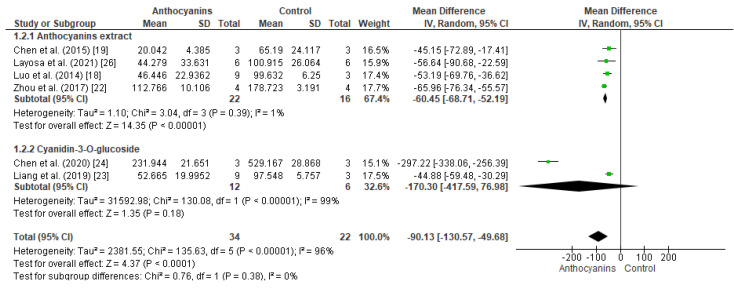
Forest plot of the meta-analysis comparing the anthocyanin-treated TNBC cell line versus the untreated controls in migration. The analysis was performed in subgroups, with four studies conducted with the anthocyanin-enriched extract and two studies evaluating isolated anthocyanin (cyanidin-3-O-glucoside). IV: inverse variance based on the mean difference; CI: confidence interval. The green box in the middle of each horizontal line represents the point estimate of the effect for each study (the larger the size of the green box, the greater the weight of the study). The black diamond represents the effect estimate of all meta-analysis studies. The position of the diamond indicates which group was significant in modulating the parameter (e.g., if the diamond is on the left side, it means that the anthocyanin group modulated the parameter. However, if the diamond touches the central vertical line, it means that there is no difference between control and anthocyanin groups) [18,19,22,23,24,26].

**Figure 5 cancers-15-02300-f005:**
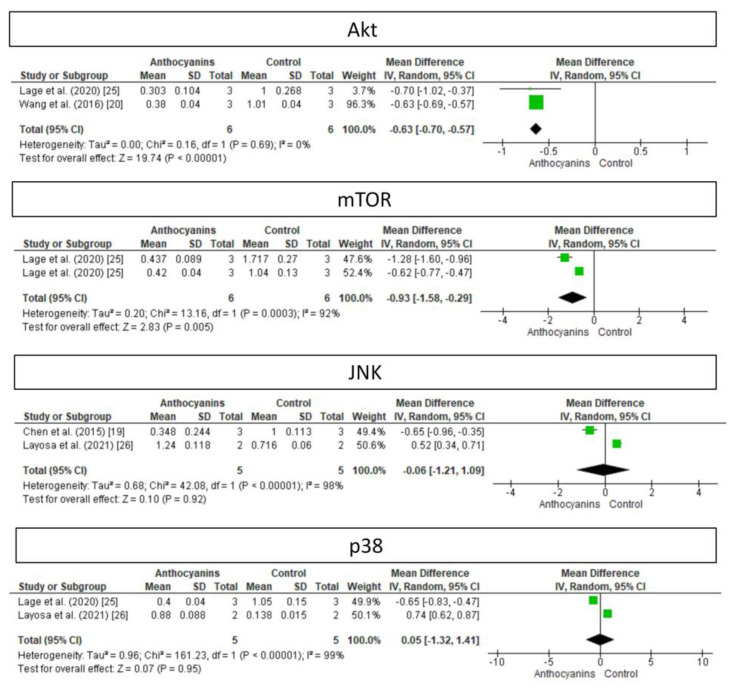
Forest plot of the meta-analysis comparing the anthocyanins TNBC cell line versus the untreated controls in migration, and in the modulation of Protein kinase B (Akt), Mammalian Target of Rapamycin (mTOR), c-Jun N-terminal kinase (JNK), and p38. IV: inverse variance based on the mean difference; CI: confidence interval. The green box in the middle of each horizontal line represents the point estimate of the effect for each study (the larger the size of the green box, the greater the weight of the study). The black diamond represents the effect estimate of all meta-analysis studies. The position of the diamond indicates which group was significant in modulating the parameter (e.g., if the diamond is on the left side, it means that the anthocyanin group modulated the parameter. However, if the diamond touches the central vertical line, it means that there is no difference between control and anthocyanin groups) [19,20,25,26].

**Figure 6 cancers-15-02300-f006:**
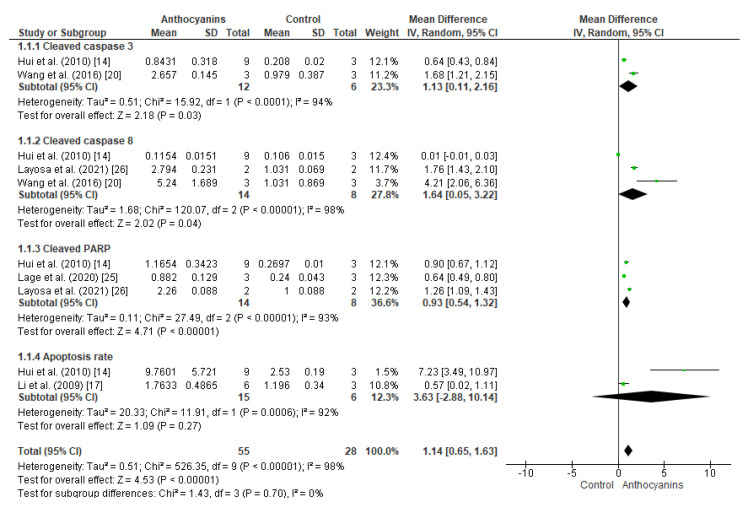
Forest plot of the meta-analysis comparing the anthocyanins TNBC cell line versus the untreated controls in apoptosis induction. IV: inverse variance based on the mean difference; CI: confidence interval. The green box in the middle of each horizontal line represents the point estimate of the effect for each study (the larger the size of the green box, the greater the weight of the study). The black diamond represents the effect estimate of all meta-analysis studies. The position of the diamond indicates which group was significant in modulating the parameter (e.g., if the diamond is on the right side, it means that the anthocyanin group modulated the parameter. However, if it touches the central line the difference between control and anthocyanin groups did not reach significance) [14,17,20,25,26].

**Table 1 cancers-15-02300-t001:** Main characteristics of the eleven studies included in the systematic review and ten included in the meta-analysis related to the role of anthocyanins in invasion, migration, modulation of cell signaling pathways, and apoptosis in TNBC cells MDA-MB-231 and HER2+ MDA-MB-453).

Author and Year	Cell Line	Treatment	Concentration	Time	Parameters	Sample Number (n)/Group	Conflict of Interest
Hui et al. (2010) [14]	MDA-MB-453	Black Rice Anthocyanins	100, 150 and 200 µg/mL	24 h	Cleaved caspase 3-8, cleaved PARP, apoptosis rate, cytochrome c, and ΔΨm	3	No
Li et al. (2009) [17]	MDA-MB-231	Eugenia jambolana Anthocyanins	100 and 200 µg/mL	48 h	Apoptosis rate	3	No information
Luo et al. (2014) [18]	MDA-MB-453	Black Rice Anthocyanins	100, 200, and 400 µg/mL	24 h	Invasion and migration	3	No
Chen et al. (2015) [19]	MDA-MB-453	Black Rice Anthocyanins	200 µg/mL	24 h	Invasion, migration, and JNK	3	No
Wang et al. (2016) [20]	MDA-MB-231	Cyanidin-3-O-glucoside	150 µM	24 h	Cleaved caspase 3 and 8, and Akt	3	No
Cho et al. (2017) [21]	MDA-MB-453	Cyanidin-3-glucoside from Mulberry	100, 200, 300, 400 and 500 µg/mL	72 h	Cleaved caspase 3, Bax and Bcl-2	Data not available	No
Zhou et al. (2017) [22]	MDA-MB-453	Black Rice Anthocyanins	200 µg/mL	24 h	Invasion and migration	4	No information
Liang et al. (2019) [23]	MDA-MB-231	Cyanidin-3-O-glucoside	5, 10 and 20 µM	24 h	Invasion and migration	3	No
Chen et al. (2020) [24]	MDA-MB-231	Cyanidin-3-O-glucoside	20 µM	24 h	Invasion and migration	3	No
Lage et al. (2020) [25]	MDA-MB-453	Dark Sweet Cherry (*Prunus avium*) Anthocyanins Enriched in Anthocyanins	70 µg C3G/mL	8 h	Cleaved PARP, mRNA levels of caspase 3-8-9 and Bax/Blc2. Akt, mTOR, and p38	3	No
Layosa et al. (2021) [26]	MDA-MB-453	Dark Sweet Cherry (*Prunus avium*) Anthocyanins Enriched in Anthocyanins	19 µg C3G/mL	48 h	Invasion, migration, cleaved caspase 8 and cleaved PARP, cytochrome c, Bax/Bcl2, JNK, and p38	6	No

## Data Availability

The data presented in this study are available on request from the corresponding author.

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
