# Peer review of "Anthocyanins Reduce Cell Invasion and Migration through Akt/mTOR Downregulation and Apoptosis Activation in Triple-Negative Breast Cancer Cells: A Systematic Review and Meta-Analysis"

_cancers, 2023, doi:10.3390/cancers15082300_

Round 1

Reviewer 1 Report (Previous Reviewer 3)

I think the authors have followed carefully the reviewers recommendations. The paper in its current form is satisfactory and suitable for publication.

Author Response

Thank you for your positive input and your feedback. We appreciate your time and effort in carefully considering our work for publication.

Reviewer 2 Report (Previous Reviewer 2)

I read this manuscript and found that authors have improved manuscript substantially. I recommend this manuscript for publication

Author Response

Thank you for taking the time to review our manuscript and for providing valuable feedback. We are pleased to hear that you found the improvements we made to be substantial and that you recommend our manuscript for publication.

Reviewer 3 Report (Previous Reviewer 4)

All my previously made comments were accepted. In my opinion updated version is ready to be published.

Author Response

Thank you for reviewing our manuscript. We greatly appreciate your thorough review and constructive feedback, which has helped to improve the quality of our work.

Reviewer 4 Report (Previous Reviewer 1)

Pls discuss and include other works of related bioactive compounds from other plants and its derivatives such as PMID: 27748367.

Author Response

Thank you for taking the time to suggest an article for us to consider. We appreciate your effort to contribute to our discussion. However, the suggested article (A Novel Resveratrol Based Tubulin Inhibitor Induces Mitotic Arrest and Activates Apoptosis in Cancer Cells) does not fit into our topic anthocyanins. Instead, we have included additional works related to anthocyanins and breast cancer such as those reporting the reduction of Akt and NF-κB in breast cancer cells after treatment with anthocyanins, and delphinidin-3-glucoside breast cancer targets (Page 10).

Paramanantham, A.; Kim, M.J.; Jung, E.J.; Kim, H.J.; Chang, S.H.; Jung, J.M.; Hong, S.C.; Shin, S.C.; Kim, G.S.; & Lee, W. S. Anthocyanins Isolated from Vitis coignetiae Pulliat Enhances Cisplatin Sensitivity in MCF-7 Human Breast Cancer Cells through Inhibition of Akt and NF-κB Activation. molecules. 2020, 25, 16:3623. doi:10.3390/molecules25163623

Yang, X.; Luo, E.; Liu, X.; Han, B.; Yu, X.; Peng, X. Delphinidin-3-glucoside suppresses breast carcinogenesis by inactivating the Akt/HOTAIR signaling pathway. BMC Cancer. 2016, 16:423. doi: 10.1186/s12885-016-2465-0.

Reviewer 5 Report (New Reviewer)

Ana Carolina Silveira Rabelo et al. in” Anthocyanins reduce cell invasion and migration through Akt/mTOR downregulation and apoptosis activation in triple-negative breast cancer cells: a systematic review and meta-analysis” show how anthocyanin enriched extracts and isolated C- 360 3-O-G were able to reduce both cell migration and invasion by mechanisms likely involving the inhibition of Akt/mTOR pathway and induction of apoptosis.

The review is well written and extremely original. Beautiful work!

Author Response

Thank you for taking the time to review our systematic review and meta-analysis. We greatly appreciate your kind words and positive feedback on our work.

Round 2

Reviewer 4 Report (Previous Reviewer 1)

The major demerit of this manuscript is lack of novelty. There are several recent review articles addressing the topic such as PMID: 27646173, PMID: 33359264 etc.

This manuscript is a resubmission of an earlier submission. The following is a list of the peer review reports and author responses from that submission.

Round 1

Reviewer 1 Report

This article reviews and analyses the use of anthocyanins in triple negative breast cancer. It is reasonably well-written, but addition of more relevant experiments are encouraged.

Pls discuss related compounds such as resveratrol and derivatives as described in PMID: 27748367. 

Author Response

Thank you for taking the time to suggest an article for us to consider. We appreciate your effort to contribute to our discussion. However, after carefully reviewing the information presented in the PMID: 27748367 article about resveratrol, we have noticed that it does not directly fit to our current topic. As such, we have decided not to incorporate this suggestion into our discussion. Thank you again for your input.

Reviewer 2 Report

The systematic review and meta-analysis is written very well. The data are very well presented and documented very well in tables and figures. The results and discussion sections are excellently presented. The English grammer mistakes/typo mistakes can be corrected.

Author Response

Thank you very much for the comment. The manuscript has been reviewed and all grammar mistakes/type were corrected.

Reviewer 3 Report

Comments to Authors

1.     In line 20, the expression should be “consists of” instead of “consists in”.

2.     In line 60 “This results in different clinical and treatment outcomes”. This statement needs to be merged with the previous sentence to empower the meaning and highlight the challenge.

3.     In line61-63, authors should Remove the abbreviations as they are misleading....they indicate the receptors after talking about the individual proteins

4.     In line 117: You described this exclusion criteria in the methods and not really in the results section.

5.     Line 126: remove “of” on apoptosis of.

6.     In table 1, the column of concentration “Does the different concentrations affect the outcomes one way or another?

7.     Authors could add the structure of anthocyanin (cyanidin-3-glucoside).

8.     In line 167-168: Now it is confusing, as you mentioned early in this paragraph that the meta-analysis was carried out in subgroups and already demonstrated that earlier and now showed a mean difference for them together. Please clarify and explain.

Reviewer 4 Report

The manuscript “Anthocyanins reduce cell invasion and migration through Akt/mTOR downregulation, but not through apoptosis activation in triple-negative breast cancer cells: a systematic review and meta-analysis” is of great interest because of the high number of breast cancer cases emerging every year and the lack of effective treatment for triple-negative variant of the disease. Ten articles were thoroughly chosen among 354 items for meta-analysis. Applied statistical methods allowed to reveal the reduction in cell migration and invasion in triple-negative breast cancer cell lines by anthocyanin but also the lack of influence on apoptosis induction was shown.

Studying of anthocyanins (and anthocyanins rich extracts) effects on different human diseases is very promising but meets many difficulties like variable anthocyanins content in different extracts applied for experiments, different cell lines or animal models etc. It leads to quite controversial conclusions and difficulties in comparison of results. So, meta-analysis like the one presented in the manuscript is very important for summarizing the data of various studies and understanding the future perspectives of anthocyanins in human disease treatment as well as possible mechanisms of anthocyanins action. Unfortunately, the most studies are focusing on human cell lines but anthocyanins action in human body and availability of biologically active anthocyanins in tumors are still of great question and concern. Despite of it the importance of the manuscript is out of the question.

There are only minor questions and suggestions.

It is better to specify Latin names for plant species like mulberry (Morus alba L.) or black rice (Oryza sativa L.)

Latin names of species should be marked Italic, like Prunus avium, Eugenia jambolana (line 145).

The same for terms like “in silico”, “in vitro”.

Perhaps, it would be more convenient for column Concentration in table 1 to mention not only concentration of anthocyanins from original articles but also some uniform units.

Author Response

  1. It is better to specify Latin names for plant species like mulberry (Morus alba L.) or black rice (Oryza sativa L.)

Answer: Species names have been added.

  1. Latin names of species should be marked Italic, like Prunus avium, Eugenia jambolana (line 145).

Answer: Latin names were mark in italic.

  1. The same for terms like “in silico”, “in vitro”.

Answer: The terms “in silico”, “in vitro” e “in vivo” were mark in italic.

  1. Perhaps, it would be more convenient for column Concentration in table 1 to mention not only concentration of anthocyanins from original articles but also some uniform units.

Answer: We agreed that unifying the units would provide a more uniform table. However, it was impossible to unify units because for the anthocyanin-enriched extract, the measurement units were “µg/mL” and “µg C3G/mL”; while “µM” were used for Cyanidin-3-O-glucoside. In the case of the “µg C3G/mL” unit, the calculations are made based on the total amount of Cyanidin-3-O-glucoside present in the enriched extract, but we cannot generalize and convert into “µg/mL” or “µM”.